# The LIM Protein Ajuba Augments Tumor Metastasis in Colon Cancer

**DOI:** 10.3390/cancers12071913

**Published:** 2020-07-15

**Authors:** Noëlle Dommann, Daniel Sánchez-Taltavull, Linda Eggs, Fabienne Birrer, Tess Brodie, Lilian Salm, Felix Alexander Baier, Michaela Medová, Magali Humbert, Mario P. Tschan, Guido Beldi, Daniel Candinas, Deborah Stroka

**Affiliations:** 1Department of Biomedical Research, Visceral and Transplantation Surgery, University of Bern, Clinic of Visceral Surgery and Medicine, Bern University Hospital, 3008 Bern, Switzerland; noelle.dommann@dbmr.unibe.ch (N.D.); daniel.sanchez@dbmr.unibe.ch (D.S.-T.); eggsl@hotmail.ch (L.E.); fabienne.birrer@dbmr.unibe.ch (F.B.); tess.brodie@dbmr.unibe.ch (T.B.); lilian.salm@insel.ch (L.S.); felix.baier@dbmr.unibe.ch (F.A.B.); guido.beldi@insel.ch (G.B.); daniel.candinas@insel.ch (D.C.); 2Department of Biomedical Research, Radiation Oncology, University of Bern, Bern University Hospital, 3008 Bern, Switzerland; michaela.medova@dbmr.unibe.ch; 3Institute of Pathology, University of Bern, 3008 Bern, Switzerland; magali.humbert@pathology.unibe.ch (M.H.); mario.tschan@pathology.unibe.ch (M.P.T.)

**Keywords:** Ajuba, metastasis, proliferation, cell differentiation

## Abstract

Colorectal cancer, along with its high potential for recurrence and metastasis, is a major health burden. Uncovering proteins and pathways required for tumor cell growth is necessary for the development of novel targeted therapies. Ajuba is a member of the LIM domain family of proteins whose expression is positively associated with numerous cancers. Our data shows that Ajuba is highly expressed in human colon cancer tissue and cell lines. Publicly available data from The Cancer Genome Atlas shows a negative correlation between survival and Ajuba expression in patients with colon cancer. To investigate its function, we transduced SW480 human colon cancer cells, with lentiviral constructs to knockdown or overexpress Ajuba protein. The transcriptome of the modified cell lines was analyzed by RNA sequencing. Among the pathways enriched in the differentially expressed genes, were cell proliferation, migration and differentiation. We confirmed our sequencing data with biological assays; cells depleted of Ajuba were less proliferative, more sensitive to irradiation, migrated less and were less efficient in colony formation. In addition, loss of Ajuba expression decreased the tumor burden in a murine model of colorectal metastasis to the liver. Taken together, our data supports that Ajuba promotes colon cancer growth, migration and metastasis and therefore is a potential candidate for targeted therapy.

## 1. Introduction

Ajuba is a LIM-domain protein which contains a unique N-terminal region, a pre-LIM region and three tandem C-terminal LIM domains [1,2,3]. LIM domains are tandem zinc-finger structures that function as a protein-binding interface and are associated with cytoskeletal organization and signal transduction from the plasma membrane to the nucleus [4]. LIM-domain proteins are highly conserved between species and Ajuba is most closely related to its family members LIMD1 and WTIP [5].

Ajuba has been found to be significantly upregulated in several cancers such as esophageal squamous cell carcinoma, cervical cancer and colorectal cancer [6,7,8,9]. In the literature, there are discrepancies as to whether Ajuba is a driver or suppressor of tumor cell proliferation. In hepatocellular carcinoma and malignant mesothelioma, Ajuba was shown to be a negative regulator of the proto-oncogene YAP and therefore was classified as a tumor suppressor [10,11]. Also, Sato et al. supported its tumor suppressor function in small cell lung cancer by demonstrating that loss of Ajuba expression resulted in enhanced tumor growth [12]. Whereas Ajuba has also been identified as a tumor promotor in cervical and colorectal cancer through positive regulation of YAP and TAZ and therefore negatively regulating the Hippo pathway [6,13].

Colorectal cancer (CRC) is one of the leading causes of cancer-related deaths, with metastasis leading to a 5-year patient survival rate of only 14% [14]. The aim of this study was to investigate the role Ajuba expression in colon cancer growth and metastasis. We found Ajuba to be highly expressed in human colon cancer and its expression is negatively correlated with patient survival. We modified Ajuba expression in the cancer cell line SW480 established from a primary Dukes’ type B colon adenocarcinoma [15] and performed high-throughput transcriptomics. Transcriptomic results were supported with mass cytometry by time-of-flight (CyTOF) and validated with biological assays. Our data demonstrates that Ajuba promotes colon cancer cell proliferation, migration and tumor metastasis in vivo.

## 2. Results

### 2.1. Ajuba Expression Is Increased in Human Colon Cancer Cell Lines and Metastatic Tumor Tissue

We compared the expression of Ajuba mRNA and protein in colon cancer tumor cells to normal tissue. Ajuba mRNA levels were significantly increased in the cancer cell lines, SW480, SW620 and HCT-116 and samples of colorectal metastasis compared to normal colon tissue (Figure 1A). Ajuba protein was clearly expressed lower in control colon tissue compared to both colon cancer cell lines and colorectal metastasis (Figure 1B). Using publicly available data from The Cancer Genome Atlas (TCGA) we compared 471 primary colon tumors to 41 healthy adjacent non-tumor tissues. Ajuba was significantly upregulated in colon cancer tissues compared to the control tissue (*p*: 7.310 × 10^−176^) (Figure 1C). Assessing its two most closely related family members WTIP and LIMD1 we found a non-significant increase (*p* = 0.570) of WTIP in colon cancer and no difference for LIMD1 (*p* = 0.401) (Figure 1C). Kaplan Meyer curves revealed that increased expression of Ajuba negatively correlated with colon cancer patient survival (*p* = 0.020) (Figure 1D). High expression of WTIP was also correlated with a significant decrease in patient survival (*p* = 0.004), whereas there was no correlation with LIMD1 expression and patient survival (*p* = 0.290) (Figure 1D). 

Ajuba overexpression could be confirmed in two patient samples of CRC metastasis in the liver using histology (Figure 1E). Colon cancer metastases were distinguished from surrounding liver tissue by H&E staining and Ki-67 which marks actively proliferating cells. Ajuba colocalized with Ki-67 expression in the metastatic cells. To summarize our findings, there is an increase of Ajuba expression in colon cancer tumors, cells lines and CRC metastasis and high protein expression correlates with poor patient survival.

### 2.2. Ajuba Is Efficiently Knocked down and Overexpressed Using Lentiviral Transduction

To study the function of Ajuba, we modulated its expression in SW480 cells using lentiviral constructs to knockdown (KD) or overexpress (OE) Ajuba protein. We selected lentiviral transduction of short hairpin (sh)RNAs due to the fact that in our hands a total knockout using CRISPR/Cas9 strategies generated non-expandable, lethal clones (Appendix A). After transduction with lentiviruses targeting a coding (shAjuba1) and non-coding (shAjuba2) region of Ajuba, we observed a significant decrease of Ajuba mRNA in shAjuba1 and shAjuba2 compared to the parental and shScrambled controls (Figure 2A). The knock down efficiency of shAjuba1 (77%) was greater than shAjuba2 (30%; Figure 2A). Transfecting the cells with an Ajuba OE construct significantly increased Ajuba mRNA levels (345%; Figure 2A). KD of Ajuba did not alter levels of its related family members, LIMD1 and WTIP (Appendix A). We confirmed Ajuba expression by immunoblot showing decreased Ajuba protein expression in shAjuba1 and 2 and increased expression in Ajuba OE (Figure 2B and Appendix A). Altered Ajuba protein expression was also observed by immunofluorescence. Ajuba protein expression was extremely low in shAjuba1 (Figure 2C, left panel), whereas there was increased Ajuba expression in OE cells with both cytoplasm and nuclear localization (Figure 2C, right panel). Interestingly, OE Ajuba protein appeared to be increased in cells that are proliferating and ready for cytokinesis as implicated by its high expression in cells with an elongated morphology (Figure 2C, arrows).

### 2.3. RNA-Seq Analysis of SW480 with Ajuba KD and OE

We next questioned how changes in Ajuba expression alter a cancer cell’s transcriptome by performing RNA-seq of SW480 with KD and OE of Ajuba. The principal component (PC) analysis of the RNA-seq data showed that the replicates of the modified cell lines clustered together (Figure 3A). The first principle component (PC 1) is correlated with the level of Ajuba, with Ajuba KD cells clustering to the left and Ajuba OE cells to the right. The second principle component (PC 2) mainly shows the differences between shAjuba1 and shAjuba2 (Figure 3A). We next assessed the number of statistically significant differentially expressed genes (DEG) in the KD and OE lines compared to their respective controls. ShAjuba1 had the highest number of DEG that were both up and down regulated (Figure 3B). A heat map of the gene expression for all significant DEG is displayed; with the technical replicates of each cell line being very similar and with close agreement between DEG expression of shAjuba1 and shAjuba2 (Figure 3C). In order to gain mechanistic insight into the effect of Ajuba KD and OE from the obtained DEG lists we performed a pathway enrichment analysis using Metascape. We identified the families of pathways that had the highest scoring adjusted *p*-values for the two KD and OE cell lines compared with their controls and showed the 20 highest scoring families of pathways (Figure 3D and Appendix A). Pathways involved in the regulation of cell differentiation, cell adhesion, Wnt signalling and epithelial cell proliferation were significantly altered by modulating Ajuba expression.

### 2.4. Pathway Enrichment Analysis Using Common DEG from the Ajuba KD Cell Lines Showed Ajuba Involvement in Cell Adhesion, Cell Differentiation and Proliferation

There were more DEG with loss of Ajuba then with gain of Ajuba expression. Therefore, we decided to conduct further analysis of the RNA sequencing data comparing the two Ajuba KD cell lines shAjuba1 and shAjuba2. There were more DEG in shAjuba1 than shAjuba2 (4927 versus 1337), consistent with the lower expression of Ajuba in shAjuba1. There were 836 DEG in common for both shAjuba1 and shAjuba2 (Figure 4A). 

Expression of the 836 genes that intersect are displayed in Figure 4B, with the yellow bar on the left of the heat map indicating all the genes upregulated in both shAjuba1 and shAjuba2, and the black bar indicating all the genes downregulated in both. The purple bar corresponds to genes with the opposite regulation. As most of the common DEG followed the same trend in both of the Ajuba KD cell lines, we can assume that they are similarly affected by the loss of Ajuba expression. To gain mechanistic insight into the DEG affected by loss of Ajuba expression, an enrichment analysis of DEG in shAjuba1 and DEG in shAjuba2 using the multiple gene list function on Metascape was done and displayed in a Circos plot. In addition to the overlap of DEG (purple lines), we also observed the genes that are not identical but are part of the same pathways (blue lines) indicating a strong functional overlap in both KD lines (Figure 4C). To further assess the functional overlap, the commonly affected pathways from the DEG of shAjuba1 and shAjuba2 were displayed in a heat map. Using multiple genes lists of shAjuba1 and shAjuba2, the 20 families of pathways that had the highest scoring adjusted p-values that are common in shAjuba1 and shAjuba2 are displayed. Some of the pathways affected by loss of Ajuba expression were regulation of cell adhesion, extracellular structure organization, cell differentiation, Wnt signalling and proliferation (Figure 4D). Those pathways were also found among the highest scoring pathways dysregulated in Ajuba OE cells (Appendix A).

### 2.5. Mass Cytometry Analysis of Ajuba KD Reveals Differential Expression of Cancer Associated Proteins

We next assessed the effect of Ajuba loss on the expression on a panel of proteins commonly associated with signaling pathways dysregulated in cancer. We employed CyTOF single cell proteomics to compare the expression of markers for metastasis, cancer stem cells and markers involved in tumor development and progression in the KD and control cells [16]. Dimensionality reduction with *t*-SNE reveals shAjuba1, shAjuba2 and the shScrambled control clustered within one sphere as they are all modified from the same parental line and show similar target expression. Nevertheless, we could observe separation of each cell line within the sphere (Figure 5A). The control localized to the right side, whereas shAjuba2 clustered in the center and shAjuba1 formed several islands in the plot. Overall the number of differentially expressed proteins from this panel was low (Appendix A). Among them was the differential expression of phosphorylated ribosomal protein S6 (pS6), an AKT effector, situated down stream of PI3 signaling pathway and known to be involved in proliferation and initiation of translation. pS6 was found to be highly expressed in the shScrambled cells (Figure 5B and Appendix A). EPHA2 expression was elevated in shAjuba1 (Figure 5C and Appendix A) and EGFR expression was elevated in shAjuba2 (Figure 5E and Appendix A). Another interesting finding was to see phosphorylated tumor suppressor TP53 (pp53) expressed more in the KD cell lines (Figure 5D and Appendix A). The significantly altered proteins supported by RNA are displayed in bar plots of the normalized values from the RNA-seq and CyTOF data (Figure 5F–I). Correlation between RNA and protein for all other proteins in the CyTOF panel are displayed in Appendix A.

### 2.6. Ajuba KD Reduces Cell Migration, Colony Formation and Metastatic Tumor Burden in Rag2−/−Gamma(c) Mice

To confirm our transcriptomic and proteomic findings that loss of Ajuba influences cell adhesion, differentiation and proliferation, we performed several biological assays in vitro and in vivo. We assessed proliferation in Ajuba KD cell lines using an MTT assay. Proliferation was significantly decreased in shAjuba1 cells whereas there was no effect in shAjuba2 cells (Figure 6A). We observed a significant decrease in colony formation after Ajuba KD in both shAjuba1 and shAjuba2 cell lines (Figure 6B and Appendix A). Furthermore, migration assay, revealed that there was a significant decrease in migration distance of shAjuba1, whereas no significant difference was observed in shAjuba2 (Figure 6C). We tested the response of the cell lines to irradiation and found that both Ajuba KD cell lines were more sensitive to irradiation compared to the control (Figure 6D).

To investigate whether loss of Ajuba expression influences cellular differentiation, we measured the stem cell marker aldehyde dehydrogenase (ALDH). In both Ajuba KD cell lines there were fewer ALDH positive cells compared to the controls (Figure 6E). To confirm the anticipated effect on metastasis, we injected the modified human SW480 colon cancer cell lines into the spleens of Rag2−/−gamma(c)−/−immunodeficient mice. With the loss of Ajuba expression there was a significant decreased tumor burden evident by fewer and smaller metastatic liver tumors (Figure 6G). This was observed directly by gross histology, counting the tumors and measuring the liver to body rate ratio (Figure 6F). Taken together our data indicate that Ajuba promotes proliferation, colony formation, migration and stem cell features in colon cancer cells. We also confirmed Ajuba promotes metastasis by a decreased tumor burden coinciding with loss of Ajuba expression in an in vivo mouse model of colon cancer metastasis to the liver.

## 3. Discussion

Using high-throughput sequencing, we provide a genome wide description of how Ajuba may be involved in pathways essential for colon cancer growth and potential for metastasis. Throughout our study, we observed stronger changes with the shRNA lentiviral construct targeting the coding region of Ajuba (shAjuba1) compared to the one targeting the non-coding region (shAjuba2). 

The differences between the two constructs was evidenced with data describing Ajuba mRNA and protein expression, outcomes in biological assays, and transcriptional changes. With RNA-seq we observed that with the more efficient KD there were a greater number of DEGs and both KD lines had significant overlap between the DEG. However, the same DEGs were not always affected in the two conditions, but many were involved in the same pathways, capturing different parts of the same biological process, thereby demonstrating robustness in the pathways identified.

Comparing the OE cell line with its control showed only few DEG. This may be explained by the fact that the parental SW480 cell line already expresses high amounts of Ajuba. Interestingly, the pathway enrichment analysis of both KD and OE lines identified similar pathways affected by altering Ajuba expression, thus strongly supporting its role in these processes. The main branches of pathways identified were proliferation, cell differentiation and epithelial-to-mesenchymal transition (EMT) which are all pathways known to be important in cancer development and metastasis. Proliferation pathways includes pathways such as epithelial cell proliferation, Wnt signaling and MAPK cascade. This coincides with a previous report that Ajuba is an important regulator of the Wnt signaling pathway [17]. In addition, there are numerous reports that Ajuba is involved in the regulation of the Hippo pathway, a signaling cascade that inhibits proliferation and promotes apoptosis and is often found to be dysregulated in cancer [5,18,19]. Finally, the cell differentiation pathways such as negative regulation of cell differentiation and epithelial cell differentiation are also well known to be dysregulated in cancer. EMT pathways include activities such as actin filament-based processes, cell projection morphogenesis, regulation of cell adhesion and wounding. Ajuba has been found to be involved in EMT and cell adhesion, a process crucial for migration and metastasis formation of tumors [8,19,20,21,22,23,24]. During EMT, cancer cells acquire more stem cell features, for example, they lose markers of differentiation and they tend to become more mobile and invasive [25]. And finally, to assess the role of Ajuba in differentiation we measured the stem cell marker ALDH [26]. We found that cells with lower Ajuba expression have fewer ALDH positive cells. This is in agreement with Lang et al. demonstrating that high NFATC2 expression enhances YAP activity and promotes stemness via upregulation of Ajuba [13].

We also investigated the effect of Ajuba KD using a single cell proteomics approach with CyTOF [16]. The *t*-SNE plots clustered according to antigen expression show a slight shift between the different samples. Many of the targets were not remarkably regulated however, the proteins that were differentially regulated were p53, pS6, EPHA2 and EGFR and correlated with the RNA sequencing results further supporting the involvement of Ajuba in these pathways. In agreement with Kalan et al., we observed that pp.53 was increased in KD cell lines [27,28]. Ribosomal protein S6 (S6) was decreased in KD cell lines. S6 is a major substrate of different protein kinases in the ribosome such as ribosomal protein S6 kinase (S6K) [29]. S6K acts downstream in the PI3 kinase pathway and its phosphorylation induces protein synthesis at the ribosome and thereby controls cell growth, proliferation and survival [30]. This supports the pathway analysis of the RNA-seq data as well as the cell proliferation biological assay that decreased Ajuba expression decreases cell proliferation. This finding however could only be shown in the more efficient KD shAjuba1 and no significant decrease of proliferation rate could be observed in shAjuba2. We hypothesized, that the remaining Ajuba expression was sufficient to maintain cell proliferation. The two receptors EPHA2 and EGFR are reported to be increased in CRC patients and have a critical role in oncogenic signaling [31,32,33]. The canonical function of EPHA2 is to inhibit cancer proliferation as well as motility whereas the non-canonical pathway EPHA2 promotes tumor survival and metastasis and drives the cells to be more dedifferentiated [34]. Another role of EPHA2 was investigated by Dohn et al. who found EPHA2 protein to be regulated by TP53 and to induce apoptosis [35]. EPHA2 was elevated in shAjuba1 cells supporting its potential role in the canonical function of EPHA2.

EGFR protects cancer cells from apoptosis, facilitate invasion and promote angiogenesis [36,37,38]. Surprisingly, we observed more EGFR in shAjuba2 cells then its control which is not congruent with our biological assays in which cells with loss of Ajuba expression have decreased migration capacities. The increase of EGFR might be due to a compensatory reaction of the cells.

Our findings agree with the current state of literature describing Ajuba as pro-proliferative in colorectal cancer [8,9,21,39]. We can state that only Ajuba and not its closely related LIM domain family members WTIP and LIMD1 are significantly increased in colon cancer compared to adjacent non-tumor tissues [8]. We demonstrate that tumor samples of CRC metastasis from the liver were highly proliferative, as shown by Ki-67 staining, and Ajuba was highly expressed in the actively proliferating cells. Nevertheless, others report Ajuba to be anti-proliferative in other cancers such as malignant mesothelioma and HCC [10,40]. The diverse functions of Ajuba are defined by its cellular localization. In the cytoplasm, the role of Ajuba is to stabilize cell junctions [41], centrosome formation [42] and to repress the Hippo Signaling pathway [5,18,19]. We also observed focal points of Ajuba in the nucleus of colon cancer cells suggesting a nuclear role of Ajuba, supporting it reported function as a transcription factor [22,24,43,44]. Ajuba does contain a nuclear export sequence and therefore can be shuttled between the nucleus and the cytoplasm [1,4,45,46]. In the nucleus, Ajuba can interact with the transcription factor SNAIL to repress E-cadherin gene expression and epithelial-to-mesenchymal transition [22,43].

In summary, Ajuba was found to be highly expressed in colorectal tumors. Its knockdown led to decreased cell proliferation, migration and colony formation and to a decreased tumor burden in a model colon cancer metastasis to the liver. Taken together, our data demonstrates the crucial role of Ajuba in driving colon cancer proliferation and its dissemination.

## 4. Materials and Methods

### 4.1. Cell Lines

Human colorectal cancer cell lines (SW480™, SW620™ and HCT-116™) were purchased from ATCC. SW480 and HCT116 are primary colon adenocarcinoma cell lines. SW480 has been classified as classified as Dukes’ B meaning that the cancer has grown through the muscle layer of the bowel. SW620 is a colon adenocarcinoma cell line isolated form a lymph node to which the primary tumor metastasized. It has been isolated from the same patient as SW480 after one year and was classified as Dukes’ C meaning that the cancer has spread to at least 1 lymph node. The cells were cultured in Dulbecco’s Modified Eagle’s Medium GlutaMAX (with 10% FBS, 100 μg/mL penicillin/streptomycin (Life Technology, Carlsbad, CA, USA) at 37 °C in a humidified incubator with 5% CO_2_. All lines have been tested and are negative for mycoplasma contamination using PCR Mycoplasma Test Kit).

### 4.2. Clinical Samples

Primary human colon tissues and CRC tumor metastases from liver were obtained from patients of the University Hospital Bern (Inselspital, Bern, Switzerland). Informed consent was obtained prior to surgery in compliance with the local ethics regulations and under approval of local ethics commission (Project-ID Nr. 2019-00157).

### 4.3. Public Data Acquisition

On the 24th of September 2019, colon cancer RNA-seq expression (counts) and survival data was downloaded from The Cancer Genome Atlas [47] specifying Primary site = Colon and Projects = TCGA-COAD.

### 4.4. Differential Expression

Differentially expressed genes from RNA-seq samples were computed with the R package DESeq2 [48]. Genes with adjusted *p*-value < 0.05 were considered statistically significant. TCGA: Primary Tumors (*n* = 471) were compared with Healthy Tissue Sample (Adjacent non-tumor tissue n = 41). For colon cancer cell lines we compared shScrambled (*n* = 2) with shAjuba1 (*n* = 3), shScrambled with shAjuba2 (*n* = 3) and Control (*n* = 3) with Ajuba OE (*n* = 3). 2 samples were discarded due to metastatic and recurrent tumors (2).

### 4.5. Survival Analysis

The survival curves were calculated with the R function survfit from the R package survival [49] with the formula Surv (time, vitalstatus)~categorie and plotted with the R function ggkm from the R package ggkm [50] with options pval = T). For patients with more than one tumor, the gene expression of the multiple tumors were averaged. The data was separated in high expression (top 20%) and low expression (bottom 80%). Using interactive tools on publicly available data visualization tools such as the Human protein Atlas the effect of different cutoff threshold can also be tested. Three samples were discarded due to missing clinical information (2) and missing days of follow up information (1).

### 4.6. Western Blot

Total protein extraction was performed using RIPA cell lysis buffer (10 mM Tris with pH 8, 1 mM EDTA pH 8, 150 mM NaCl, 0.5% NP40) with addition of protease inhibitors (1 mM NaF, 10 mM NaVO_3_, 1 mM PMSD, 1X protease inhibitor cocktail—P1860, Sigma, (St. Louis, MO, USA). The cell lysates were sonicated (Sonopuls, Bandelin, Berlin, Germany) then centrifuged. Snap frozen tissue pieces were dissociated using a TissueLyser (Qiagen, Hilden, Germany) for 2 min at 20 Hz in RIPA buffer. The protein lysate concentrations were determined with the Bio-Rad Protein Assay System (Bio-Rad, Hercules, CA, USA) as described by the manufacturer. Equal amounts of proteins were separated by SDS–PAGE and transferred onto a nitrocellulose membrane using the iBlot2 Gel transfer device. The membrane then was blocked in 5% non-fat dry milk dissolved in PBS for 1 h followed by incubation with the primary antibody overnight at 4 °C. After incubation with the HRP conjugated secondary antibody, chemiluminescent reaction was performed with Western Lightning Plus-ECL from Perkin Elmer ((Waltham, MA, USA). Membranes were developed using the x-ray film processor Curix 60 (AGFA, Mortsel, Belgium). The band size was estimated using Page Ruler™ Prestained Protein Ladder (Fermentas, Waltham, MA, USA) and Precision Plus Protein™ DUAL Color Standards (Bio-Rad, Hercules, CA, USA #161-0374). Primary antibodies used were rabbit monoclonal anti-Ajuba (1:1000 dilution, Cell Signalling Danvers, MA, USA) and HRP-conjugated secondary antibodies used were goat anti-rabbit (Dako, Glostrup, Denmark). β-actin-HRP (1:100,000 dilution, Sigma, St. Louis, MO, USA) was used as a loading control.

### 4.7. Quantitative Real-Time Reverse Transcription PCR

Total RNA was isolated from human samples and cell lines using NucleoZOL (Macherey-Nagel, Macherey-Nagel, Dürren, Germany) according to manufacturer’s protocol. The quality and concentration of RNA were measured using Nanodrop 2000 Spectrophotometer Thermo Scientific, Waltham, MA, USA). Five hundred ng of total RNA was used for cDNA synthesis using Omniscript RT Kit 200 (Qiagen, Hilden, Germany). mRNA was analysed by quantitative RT–PCR with TaqMan gene expression assays and reagents according to the standard protocols (Applied Biosystems, Foster City, CA, USA). using specific primers and housekeeping genes 18S FAM as control. We used the TaqMAN ViiA TM 7 Real-time PCR system from Applied BioSystems for the amplification steps and data collection. Log 2-fold changes were computed using the ΔΔCt method. Ct values of target genes (TG) were calculated relative to a reference gene (RG, 18S) using the following formula: ΔCtTG = CtTG − CtRG. Experimental groups (TG) are normalized to control group (CG): ΔΔCt = ΔCtTG − ΔCtCG, and fold increase = 2^−ΔΔCt^.

### 4.8. Lentiviral Transduction

Due to the fact that the Ajuba KO cells were not viable we decided to use shRNA to knockdown and overexpress Ajuba. The CRC cell line SW480 was transduced with two independent shRNAs targeting Ajuba and one lentiviral Ajuba OE construct. All experiments were carried out on cells at 25–50% confluence. Cells were transfected with lentiviral supernatant, containing shRNA targeting Ajuba or Ajuba OE construct in DMEM with 10% FBS. The viral supernatant was added onto the cultured cells in a total volume of 900 μL and incubated for 3 h then centrifuged at 1400 rpm for 5 min, incubated for an additional 3 h, then complete medium was added to a final volume of 2 mL and incubated for 48 h. shRNA lentiviral constructs were purchased from MISSION^©^ ((Sigma, St. Louis, MO, USA).

Clone shAjuba1: targeting the coding region NM-032876.4-1385s1c1Clone shAjuba2: targeting the non-coding region 2URT NM_032876.4-2786s1c1Clone shScrambled: pLKO.1-puro Non-Mammalian shRNA control plasmidTo overexpress Ajuba, SW480 cells were transduced with lentivirus containing an Ajuba overexpressing constructClone Ajuba OE Harvard PlasmID:Phage_CMV_C_FLAG_HA_IRES_PURO

As a control for the Ajuba overexpressing cells, SW480 cells were transduced with the Ajuba overexpressing construct, in which we clonally excised the exogenous Ajuba sequence. Stable cell lines were positively selected using 1.5 μg/mL puromycin (Life Technologies, Carlsbad, CA, USA). The efficiency of the transduction was assessed by real-time qPCR and immunoblot.

### 4.9. Immunofluorescence

Human colon and colon cancer metastases from the liver were OCT preserved and 4μm cryosections were cut. The cryocut sections or cultured CRC cell lines were fixed using 4% paraformaldehyde for 10 min at room temperature. Cells were then permeabilized and blocked with 0.2% Triton-X and 5% goat serum in PBS for 20 min. After that monoclonal Ajuba ab (1:200 dilution, Cell Signalling, Danvers, MA, USA) was used and incubated over night at 4 °C. After three washing steps with PBS, 0.25%BSA and 0.1% Triton-X, secondary antibody (anti-rabbit conjugated to cy5 (1:1000, Life Technologies, Carlsbad, CA, USA), was incubated for 1 h at room temperature. Finally, nuclei were stained with DAPI (1:2000, Cell Signalling, Danvers, MA, USA) for 30 min at room temperature. Immunohistochemistry was imaged using fluorescence microscopy (LCI DMI4000 B, Leica, Wetzlar, Germany).

### 4.10. RNA-Sequencing

RNA was isolated from SW480 cells using the ReliaPrep RNA cell Miniprep System kit (Promega, Madison, WI, USA). The quality and concentration were measured with a 2100 BioAnalyzer (Agilent, Santa Clara, CA, USA). The isolated RNA was then sent for sequencing according to the following parameters: paired-end with reads of 50 bp, TruSeq Stranded mRNA. The RNA was sequenced with a NovaSeq6000 (Illumina, San Diego, CA, USA).

### 4.11. Alignment

FASTQ files were aligned to the human reference genome hg38 with HISAT2 [51]. Resulting sam files were transformed into bam with SAMtools [52]. The reads were counted with the R function featureCounts from the R package Rsubread [53] with options isPairedEnd = TRUE, GTF.featureType = “exon” and GTF.attrType = “gene_id”.

### 4.12. Data Visualization

Data was transformed to reads per million (RPM) for visualization. Principal component analysis was done with the R function prcomp on the log(1+x) transformed data. Heatmaps were done with R function heatmap.2 from the R package gplots [54] with average linkage and Pearson distance; for gene expression, data was scaled from 0 to 1.

### 4.13. Venn Diagram

Venn diagrams were drawn using custom Venn diagrams from Bioinformatics & Evolutionary Genomics web tools [55].

### 4.14. Pathway Enrichment Analysis

Pathway enrichment analysis was performed using Metascape Gene Annotation & Analysis Resource [56]. We used the significantly differentially expressed genes looking at the two Ajuba KD compared with the control, shScrambled and Ajuba OE compared with its control. For shAjuba1 we restricted the enrichment analysis to the top 2500 DEG sorted by *p*-value. For multiple gene lists, we used the Metascape multiple gene list function. The thresholds applied to select for the DEG genes was adjusted *p*-value below 10^−10^ for shAjuba1 and adjusted *p*-value below 0.003 for shAjuba2. The top 20 statistically significant family of pathways, clustered by Metascape, were displayed. The DEG were also displayed as a circos plot in order to show genes that are common or are part of the same pathways. The gene list of the selected pathways were obtained with the R function gconvert from the R package gProfileR2 [57]. And displayed as stackplots showing the fraction of upregulated and downregulated genes, the total number of DEG and the significance of the pathway.

### 4.15. Mass Cytometry by Time-of-Flight (CyTOF)

A total of two million cells per cell line were used for CyTOF analysis. The cells were stained with cisplatin to identify live cells and incubated for 10 min at RT. The samples were then fixed, permeabilized and barcoded using the Pd 20-plex barcoding kit (Fluidigm, San Francisco, CA, USA) according to manufacturer’s protocol. The samples were pooled in one tube and stained with metal-conjugated cell surface antibodies according to previously established titrations, cells were then fixed and permeabilized with the FoxP3 intranuclear staining kit as directed in the kit and intracellularly and intranuclearly stained with the targets in these compartments. Lastly, cells are placed in DNA intercalation solution (iridium in Fix and Perm buffer) overnight at 4 °C. The following morning cells were washed 3 times to remove salts and proteins and acquired on the Helios mass cytometer in Maxpar water containing 4-element beads. A minimum of 150,000 cells per cell line was recorded.

### 4.16. Mass Cytometry Analysis

The raw fcs file was normalized with the R function normalizer_GUI from the R package premessa [58] and debarcoded with the R function debarcoder_GUI from premessa. The resulting files were gated in Cytobank to discard beads, dead cells, pressure spikes over time and doublets. The cleaned files were analyzed with the R as follows: files were first read as a flowFrame object with the package flowCore [59]. To visualize the data, 5000 cells from each sample were randomly selected, their expression was arcsinh transformed, and we performed a *t*-SNE dimensionality reduction the with R function Rtsne from the R package Rtsne [60].

### 4.17. Comparing RNA-Seq with Mass Cytometry Results

In order to compare the mRNA and the protein expression, first we averaged the RPM values of the RNA-seq among the replicantes. Second, we normalized the RPMs values and the arcsinh transformed CyTOF data to range in [0,1] by dividing by the maximum observed value in Scrambled, sh1 and sh2, or equivalently, by applying function f(x) = x/max(x), independently for RNA-seq and CyTOF. The rescaled values where represented in barplots.

### 4.18. Cell Proliferation Assay

Cells were seeded in a 96-well plate at a density of 2000 cells per well in 200 μL of medium. Cells were incubated for at least 4 h to attached to the plate. Every day at the same time, MTT (5 mg of thiazolyl blue dissolved in 5 mL DMEM) was added in one tenth of the original culture volume (20 μL for 200 um plated) in each well (4 wells per time point and condition) and incubated for 1 h. The medium then was discarded and replaced by 200 μL of DMSO. Using an Infinite 2000 (Tecan, Männedorf, Switzerland) the plate was shaken and read at an absorbance of 570 nm.

### 4.19. Colony Formation Assay

One thousand cells per well were platted in 6-well plates and incubated for 7 days. To stop the colony formation, the medium was removed and cells were washed twice with DPBS before being dried. Crystal violet was used to stain the cells (3 g crystal violet, 99.9 mL methanol, 49.9 mL acetic acid) by incubating each well for 30 min at room temperature. The number of colonies were counted using the Colcount (Oxford Optromix, Abingdon, UK).

### 4.20. Migration Assay

Migration distance was assessed using silicon stoppers in a 96-well plate, plated with a density of 50,000 cells in 200 μL medium per well. Cells were plated in quadruplicates and were left to adhere for 6 h with the silicon stoppers. Afterwards, the stoppers were carefully removed and pictures were taken at 0 h and 24 h under 4× magnification using a light microscope. The radius of the silicon stopper area was calculated by measuring the cell free area using ImageJ package. The migration distance was finally computed by calculating the difference in radius at the different time points in micrometres (μm).

### 4.21. Irradiation

In 6-well plates, cells were platted with a density of 1000 cells in 200 uL medium per well. After letting the cells adhere for 6 h, the plates underwent irradiation in a Gammcell 40 (Best Theratronics, Ottawa, Canada) at the following doses: 0, 2, 3, 4, 5, 6 and 8 Grays. Non-irradiated cells were used as a control. The sensitivity after irradiation was assessed by the cells capacity to form colonies as described under Colony formation assay.

### 4.22. ALDH Assay

Two Million cells were seeded into 10 cm diameter dishes and let to adhere overnight. The samples were processed according to protocol using the Aldefluor^TM^ kit from Stemcell Technologies (Vancouver, BC, Canada). The samples were analysed using FACS (LSR II, BD Biosciences, Franklin Lakes, NJ, USA) recording a total number of 10,000 events. Data was analysed and gated using FlowJoTM 10 in order to remove debris, and duplet cells. Finally, gates were set according to the negative control of the cell lines where DEAB was added. The gates are set to contain exactly 3% positive cells in the negative control and gates were kept the same for the same cell line in order to assess the percentage of cells that have shifted from the negative population.

### 4.23. In Vivo Metastases

RAG 2−/−y chain KO mice were injected with 1 million of SW480 cells line either shScrambled, shAjuba1 or shAjuba2 in order to assess metastatic behavior of the different cell lines. All surgical procedures were performed under laminar flow and under sterile conditions using a general anaesthesia with intraperitoneal injection of fentanyl, midazolam and medetomidin. Anesthetized mice were immobilized in a supine position and the abdomen was entered through a midline incision. After exposure of the spleen, the cells were injected in a total volume of 100 μL in PBS directly into the spleen, similarly as previously described by Soares et al. [61]. After injection, a cotton swap was applied of the place of injection for 30 s in order to avoid reflux of the cells. The abdomen was closed with a two-layer running suture. Tumor formation in the spleen and metastatic development the liver, were visually detected at 7 weeks post-operation. Mice were sacrificed using intraperitoneal injection of terminal anaesthesia and organs harvested weighed and photographed. Samples of liver and spleen were snap frozen and paraformaldehyde conserved for further histological, mRNA and protein analysis.

### 4.24. Graphs and Statistical Analysis

R version 3.5.1 was used for displaying and computation of publicly available data, RNA-seq and CyTOF graphs using the R-packages ggplot2 [62]. The graphs and the statistics for qPCR, and in Figure 6 were done by using GraphPad Prism software (San Diego, CA, USA). *p*-values were calculated using an unpaired, two-tailed Student’s *t*-test or two-way ANOVA with no repeated measures and Tukey adjusted for multiple comparison. For all analyses NS denotes *p* > 0.05, * *p* < 0.05, ** *p* < 0.01, *** *p* < 0.001, **** *p* < 0.0001.

### 4.25. Data Availability

All data are available on Genome Expression Omnibus repository with the GEO accession number GSE147111.

## 5. Conclusions

Here, we show that Ajuba expression in colorectal tumors is associated with a significant decrease in patient survival. We give evidence for the role of Ajuba in proliferation, EMT, resistance to therapy and demonstrate the importance of Ajuba in colon cancer metastasis. Taken together, our data suggests Ajuba supports tumor growth and metastasis in CRC and as such, warrants further investigation as a possible biomarker and therapeutic target.

## Figures and Tables

**Figure 1 cancers-12-01913-f001:**
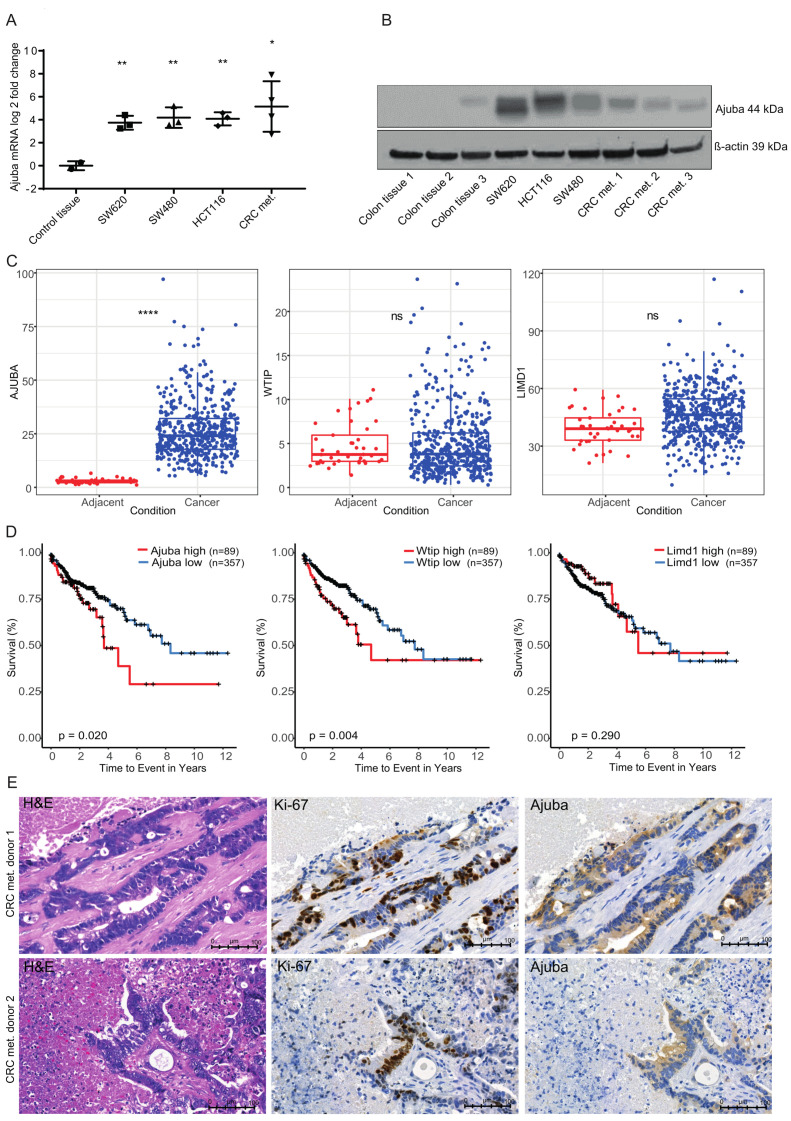
Ajuba expression is increased in human colon cancer cell lines and metastatic tumor tissue. (**A**) Ajuba mRNA expression level measured by RT-qPCR in colon cancer cell lines and human tissue of CRC metastasis in the liver normalized to human control Intestine. Control tissue vs. SW620 = 0.0048, Control tissue vs. SW480 = 0.0093, Control tissue vs. HCT116 = 0.0033, Control vs. CRC metastasis 0.0360. (**B**) Western blot analysis of Ajuba protein in human primary colon, colon cancer cell lines and human tissue of CRC metastasis in liver. (**C**) Ajuba, WTIP and LIMD1 normalized mRNA expression in (471) tumor and adjacent non-tumor tissues (41) from The Cancer Genome Atlas (TCGA). Ajuba vs. control tissue *p* = 7.310 × 10^−176^, WTIP vs. control tissue *p* = 0.570, LIMD1 vs. control tissue *p* = 0.401. (**D**) Kaplan-Meier survival curves used to calculate the colon cancer patient survival according to their Ajuba, WTIP and LIMD1 expression. The data was separated into high expression (top 20% *n* = 89) and low expression (bottom 80% *n* = 357). Increased expression is negatively correlated with survival, Ajuba *p* = 0.020, WTIP *p* = 0.004, LIMD1 p = 0.290. (**E**) Human CRC metastases in the liver from two patients, stained for H&E, Ki-67 and Ajuba. Ajuba is efficiently knocked down and overexpressed using lentiviral transduction. For all analyses NS denotes *p* > 0.05, * *p* < 0.05, ** *p* < 0.01, *** *p* < 0.001, **** *p* < 0.0001.

**Figure 2 cancers-12-01913-f002:**
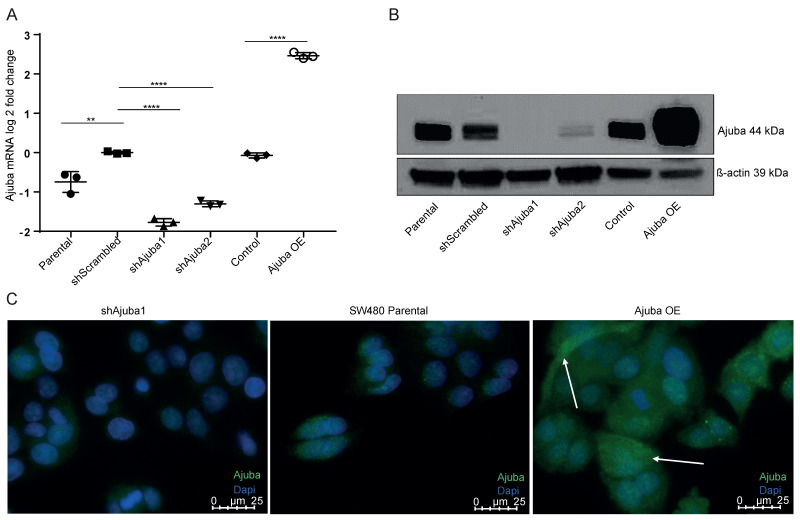
Ajuba was efficiently knocked down and overexpressed using lentiviral transduction. (**A**) Ajuba mRNA expression level measured by RT-qPCR in colon cancer cell line SW480 where Ajuba has been previously transfected with lentivirus (*n* = 3, one representative transfection shown). ShScrambled vs. Parental = 0.0083, shScrambled vs. shAjuba1 ≤ 0.0001, shScrambled vs. shAjuba2 ≤ 0.0001, Control vs. Ajuba OE ≤ 0.0001. (**B**) Western blot analysis of Ajuba protein in colon cancer cell lines where Ajuba has been KD and OE. (**C**) Immunofluorescence pictures of SW480 CRC cell line shAjuba1, parental and Ajuba OE stained for Ajuba and DAPI. The upper arrow indicates a cell with elongated morphology and the lower arrow indicates a cell that is ready for cytokinesis (40× images). For all analyses NS denotes *p* > 0.05, * *p* < 0.05, ** *p* < 0.01, *** *p* < 0.001, **** *p* < 0.0001.

**Figure 3 cancers-12-01913-f003:**
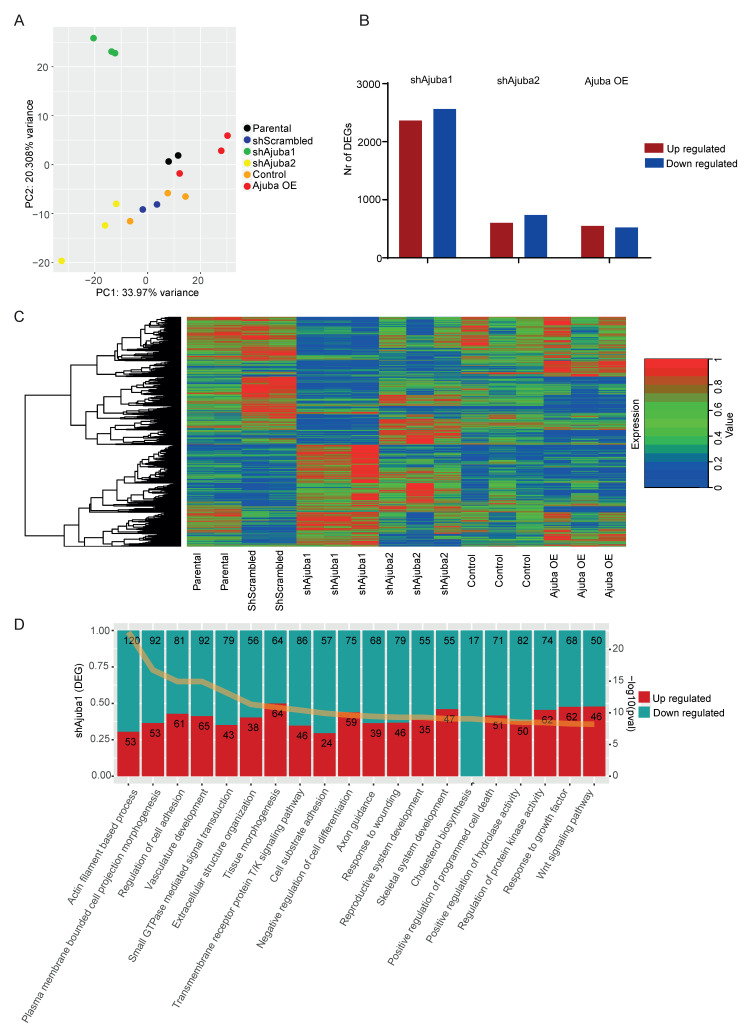
RNA-seq analysis shows most DEGs between shAjuba1 and its control. (**A**) Principle component analysis of RNA-sequencing of all conditions of SW480 colon cancer cell lines. (**B**) Number of differentially expressed genes that are up or down regulated between the different samples and their respective control (shAjuba1 and shAjuba2 compared with shScrambled and Ajuba OE compared with control). (**C**) Heat map of expression level of all significantly differentially expressed genes. (**D**) Pathway enrichment analysis of significantly differentially expressed genes between ShAjuba1 and its shScrambled control using Metascape analysis. The yellow line across the figure represents the calculated *p* value.

**Figure 4 cancers-12-01913-f004:**
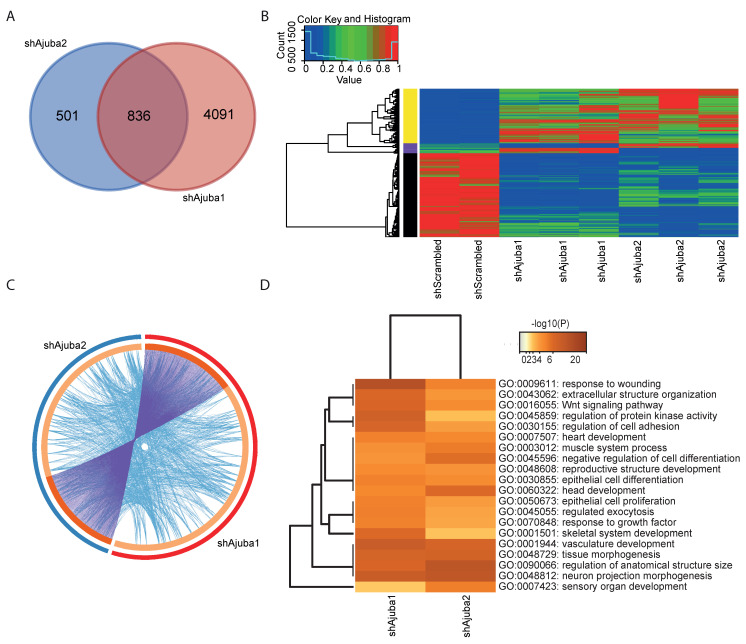
Common DEG of the two Ajuba KD are involved cell adhesion, cell differentiation and proliferation. (**A**) Venn diagram of commonly expressed genes comparing all DEG of shAjuba1 and its control and shAjuba2 with its control. (**B**) Heat map of all 836 commonly altered DEG from shAjuba1 and shAjuba2. (**C**) Circos plot calculated using multiple gene list on Metascape shows how the genes from shAjuba1 and shAjuba2 overlap and in which pathways they are involved. The blue line on the outside corresponds to the shAjuba2 gene list and the red line on the outside to shAjuba1. On the inside, Dark orange color represents the genes that are common in shAjuba1 and shAjuba2, the light orange color represents genes that are uniquely expressed by one condition. The purple lines are linking the exact genes that are common in shAjuba1 and shAjuba2. Blue lines link genes that are different but have the same gene ontology term. (**D**) Heat map showing the top 20 statistically significant family of pathways, clustered by Metascape, were displayed showing commonly altered pathways from shAjuba1 and shAjuba2 the yellow line showing their *p*-value and in which gene ontology term they are involved.

**Figure 5 cancers-12-01913-f005:**
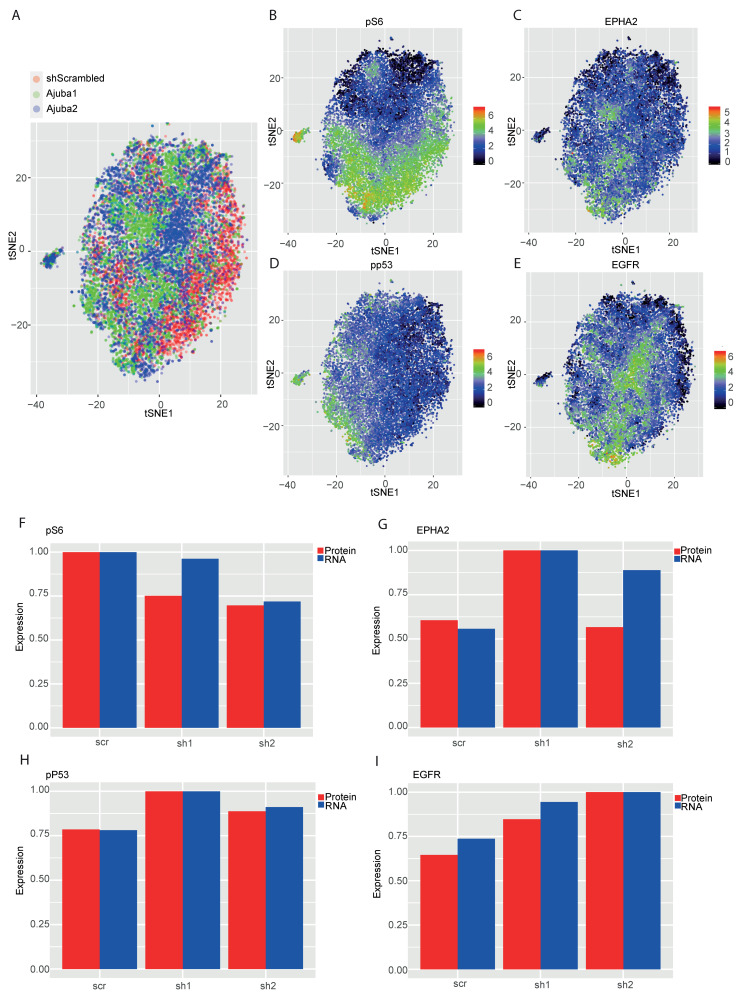
Mass cytometry analysis of Ajuba KD reveals differential expression of cancer associated proteins. (**A**) The *t*-SNE plots are clustered according to their barcoding and coloured according to the sample they come from. (**B**–**E**) The *t*-SNE plots are clustered according to their antigen expression and coloured according to a specific ab expression. All graphs have been displayed using function Rtsne from the R package Rtsne. (**F**–**I**) Bar graphs displaying correlation between protein and RNA expression. Averaged and normalized RPM values of the RNA-seq and the arcsinh transformed CyTOF data were computed independently and the rescaled values used to compare expression.

**Figure 6 cancers-12-01913-f006:**
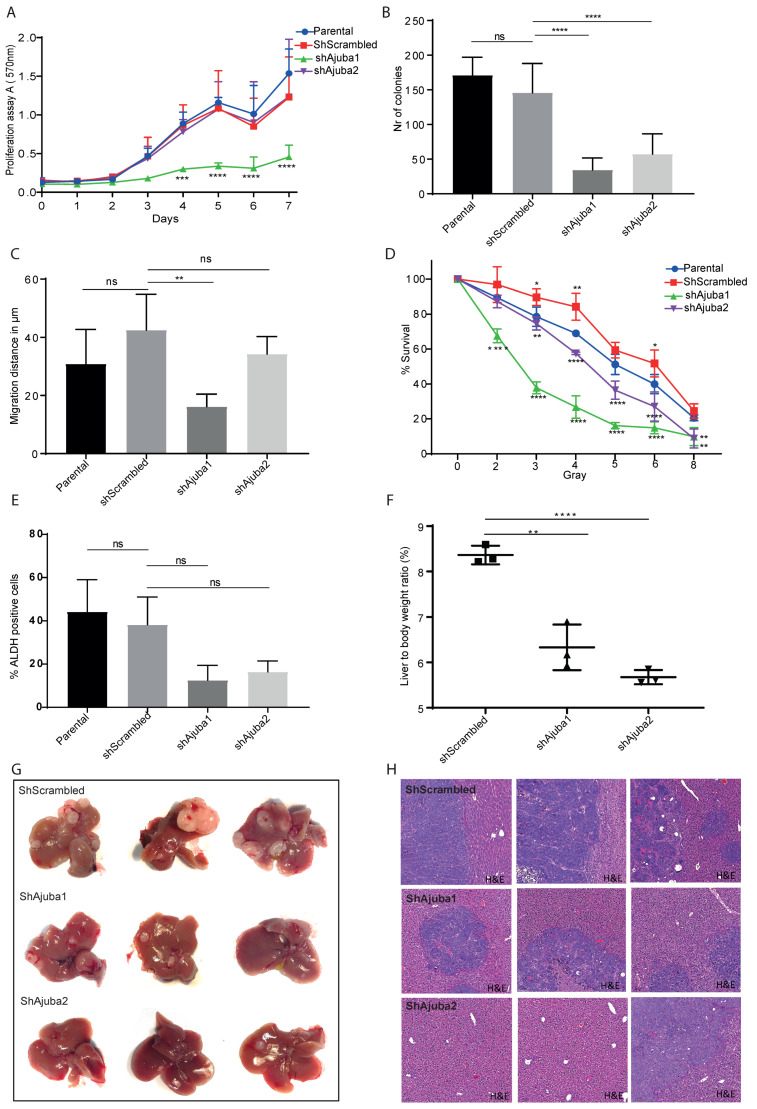
Ajuba KD reduces cell migration, colony formation and metastatic tumor burden in Rag2-/- gamma(c) mice. (**A**) Proliferation assay measuring the absorbance at 570 nm after addition of MTT over the span of one week. Statistics were computed with Graphpad performing a two-way Anova with Tukey corrections for multiple comparisons (*n* = 3 a combination of all 3 independent experiments are displayed here). Day 4: Shscrambled vs. shAjuba1 = 0.0008, Day 5: Shscrambled vs. shAjuba1 ≤ 0.0001, Day 6: Shscrambled vs. shAjuba1 ≤ 0.0001, Day 7: Shscrambled vs. shAjuba1 ≤ 0.0001. (**B**) Graph displaying number of colonies for each condition. The colonies are stained with crystal violet and counted with ColCounter. Parental vs. shScrambled = 0.1410, Shscrambled vs. ShAjuba1 ≤ 0.0001, shScrambled vs. shAjuba2 ≤ 0.0001. (**C**) Migration assay showing difference of the radius in μm, calculated between day 0 and day 1 of the migration. Measurement of areas was done by ImageJ. Parental vs. shScrambled = 0.1471, Shscrambled vs. ShAjuba1 = 0.0014, shScrambled vs. shAjuba2 = 0.1724. (**D**) Percentage of colonies that survived after being irradiated at 2, 3, 4, 5, 6 and 8 grays. Colonies were counted with the ColCounter. Statistics was computed with Graphpad performing a two-way Anova with Tukey corrections for multiple comparisons. 2 Grey: shScrambled vs. shAjuba1 ≤ 0.0001; 3 Grey: shScrambled vs. Parental = 0.0365, shScrambled vs. shAjuba1 ≤ 0.0001, shScrambled vs. shAJuba 2 = 0.0024; 4 Grey: shScrambled vs. Parental = 0.0020, shScrambled vs. shAjuba1 ≤ 0.0001, shScrambled vs. shAJuba 2 ≤ 0.0001; 5 Grey: shScrambled vs. shAjuba1 ≤ 0.0001, shScrambled vs. shAJuba 2 ≤ 0.0001; 6 Grey: shScrambled vs. Parental = 0.0236, shScrambled vs. shAjuba1 ≤ 0.0001, shScrambled vs. shAJuba 2 ≤ 0.0001; 8 Grey: shScrambled vs. shAjuba1 = 0.0032, shScrambled vs. shAJuba 2 = 0.0013. (**E**) Graph displaying percentages of ALDH positive cells using Aldefluor kit from Stemcell Technologies. Cells were gated as negative according to the internal control where samples was treated with DEAB (Appendix A). Parental vs. shScrambled = 0.6210, Shscrambled vs. ShAjuba1 = 0.0883, shScrambled vs. shAjuba2 = 0.1177. (**F**) Liver to body weight ratio (%). Shscrambled vs. ShAjuba1 = 0.0029, shScrambled vs. shAjuba2 ≤ 0.0001. (**G**) pictures of livers and (**H**) H&E staining of Rag2-/-gamma(c)-/-immunodeficient mice 7 weeks after intraspleenic injection of colon cancer cell lines. For all analyses NS denotes *p* > 0.05, * *p* < 0.05, ** *p* < 0.01, *** *p* < 0.001, **** *p* < 0.0001.

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
