# Peer review of "The LIM Protein Ajuba Augments Tumor Metastasis in Colon Cancer"

_cancers, 2020, doi:10.3390/cancers12071913_

Round 1
Reviewer 1 Report
The authors have addressed all issues I raised.
Some journal editing required - with changes in figures there are blank spaces in teh PDF version.
Author Response
Dear Reviewer,
Thank you for reviewing our manuscript. All the typos have been adressed and english was checked by two native english speaking scientists.
Reviewer 2 Report
Although I received your e-mail and read it at home early this morning, I have found that somehow the e-mail text has been lost from my computer of my office. Because the attached files have been retrievable, I have read through them. Here’s my comment: The authors have addressed all the comments in an appropriate manner, at least for the parts of Reviewer 1. Hope this evaluation is helpful for you.Author Response
Dear Reviewer,
Thank you for reviewing our manuscript.
This manuscript is a resubmission of an earlier submission. The following is a list of the peer review reports and author responses from that submission.
Round 1
Reviewer 1 Report
In this study, Dommann et al. described the role of Ajuba in colon cancers. The authors found that Ajuba is overexpressed in colon cancer tissues and its overexpression is correlated with poor prognosis. Carcinogenesis-related pathways were identified by RNA-seq and CyTOF in KD and/or OE cells. In vitro biological assays had been performed to assess the possible function of Ajuba in KD colon cancer cells, there are some major issues that need to be addressed.
The organization of the figures and the figure legends in the supplementary files are extremely rough. Figure legends do not match the Figures after Figure S2. For example, Figure legend of S2 described the result for Fig S3 and the legend of S3 is for Fig S2. Figure S4 showed data of CyTOF while the legend was for RNA-seq. After Fig S4, the figure sequence is incorrect and there is no Figure S7 in that file but only the legend. All of the supplementary data should be verified and corrected carefully. The authors mentioned in the legend of Fig S7 (which supposed to be the legend of Fig S6) that Figure 1B and Figure S6B were derived from the original WB data showed in Figure S6A. However, the sequence of sample labeling in Figure 1B and S6B is not consistent with the hand writing labels in the b-actin panel of S6A which needs to be corrected. In Figure 1D, the case number of low/high expression of each gene should be indicated in the survival data. Figure 1D. The authors divided the high and low expression groups by top 20% and bottom 80% of gene expression. The authors should explain the rationale and why the percentage/cut-off value was determined that way. The authors showed the pathway enrichment and heat map analysis of the affected pathways of KD and OE cell lines by RNA-seq (Fig 3,S2,S3). The authors need to list the upregulated/downregulated genes of the RNA-seq results (for example, the 835 commonly altered DEG). Although the legend of Fig S5 seemed to correlate with the result of Fig S4, the description in the legend is problematic. The legend indicated that both KD and OE results were showed while in fact only KD data is revealed. Figure S4. There is no “E” panel in this fig, only C, D, F, and G. The authors took efforts to do the RNA-seq and CyTOF analysis of the Ajuba-KD cells. How many altered genes were found consistent between the two data sets of the RNA-seq and CyTOF? If discrepancy occurred, what is the plausible reason? Although the knock-down of Ajuba by shRNA was more efficient in shAjuba1 than in shAjuba2 KD cells by RT-qPCR (Fig 2A), the protein expression of Ajuba seemed to be totally vanished in both KD cells, as revealed by the WB of Fig 2B and Fig S6C. However, these two cells behaved very differently in the biological assays (results of Fig 6). What could be the underlying cause of such difference since the protein was nearly equally with the knock-off? The classical cell cycle analysis by propidium iodide staining and flow cytometry is not able to differentiate G0 from G1 cells since they contain the same amount of DNA content. Usually in a PI-based cell cycle assay, the cells are divided into G0/G1, S, G2/M, and in some occasion, a sub-G1 group if the cells are disintegrated. How do the authors calculate the separated “G0” cells in the Fig 6B simply by merely the PI staining procedure as described in the methods? In addition, the authors claimed that an increased number of cells in G2 cells was observed in shAjuba1 cells (Fig 6B). However, the percentage of the increase seemed relatively small and non-significant comparing to that of the scrambled cells. What is the exact percentage of cells in each group (G0/G1-S-G2/M) of the figure? The authors observed a more mesenchymal cell shape in shAjuba1 KD cells while ShScrambled and shAjuba2 cells both displayed a more epithelial cell morphology (Figure 6F). This observation conflicts with the following ALDH assay revealing that both shAjuba cells, but not the ShScrambled cell, had lower amount of ALDH activity (and hence, lower stemness; Fig 6G). This also leads to the inconsistency of the conclusion made at the last sentence of the abstract “Our data indicates that increased Ajuba expression observed in colon cancer supports epithelial-to-mesenchymal transition” while in fact the shAjuba1 KD cell showed mesenchymal characteristics. The authors had also constructed an Ajuba overexpression SW480 cells in this study. It would be more convincing if the authors include this ectopicly overexpressed cell line in these biological assays. There are some grammatical errors across the manuscript (lack of verb or conjunction in sentences). In addition, some sentences seem to be incorrectly inserted (For example, Page 10, line 253, the last sentence of the 3rd paragraph of discussion). The authors need Professional English correction service agency to check their manuscript.
Reviewer 2 Report
Fig. 1D) please add the unit of time (Years?) to the label of the axis of abscissae.
line 111 (page 4): delete the redundant "a" at the beginning of the line.
line 144 (page 6) correct "verses" to "versus"
Reviewer 3 Report
Based on published data that the cytoskeletal protein Ajuba has ascribed oncogenic roles in small-cell lung cancer cells, Domman et al. determined whether this was the case in colon cancer. Domman et al. used compiled expression profiles derived from TCGA data, cell lines and their own clinical samples to establish that Ajuba is overexpressed in colon cancer cells. The study went on to establish functional data that Ajuba contributes functional oncogenic properties to colon cancer cell lines using loss-of-function studies. The loss-of-function studies were carried out using two siRNAs that give different phenotypic results in terms of altering gene expression profiles, proliferation, cell migration etc…
The lion’s share of the data generated in the study was essentially a comparison of the phenotype data generated using the two shRNAs and drawing conclusions based on the common properties endowed by their expression. Accordingly, the primary issue with the study is the lack of an additional loss-of-function technique that resolves the ‘on-target effects’ of shRNA expression – CRISPR/Cas9, dominant negative expression… – this is a requirement for publication of the study.
A second issue with the study is the conclusion that Ajuba expression impacts on metastasis, as mentioned in the title. This was never tested directly and the conclusion that metastasis is facilitated by Ajuba overexpression rests solely on correlation – the fact that there is a gene signature that can be drawn out of the extensive expression changes with Ajuba depletion. Data obtained with the silicon stopper assay using cell lines does not support this conclusion.
Less Major issues:
Why are the gates different amongst sample sets for the ALDH-assays?
Expression studies carried out by the authors employed a very limited clinical sample set (1-2 clinical samples). A more extensive clinical dataset would be convincing.
It is unclear why the authors chose small-hairpins versus CRISPR/Cas9 as a loss-of-function method to examine Ajuba function - although shRNA offers the benefit of an inducible system – why wasn’t this used?
No explanation is provided as to why shRNA1 decreases proliferation whereas shRNA2 does not as shown by the growth curves. This does not appear to be due to effects on the cell cycle. Could effects on apoptosis or senescence explain the differences?
What is the point of the CyTOF experiments? How does this data explain the observed phenotypic differences with Ajuba depletion.
Minor issue
Supplementary figures: S2 and S3 legends have been mixed up.